# Production of a monolithic fuel cell stack with high power density

Stéven Pirou [1]✉, Belma Talic [1,2], Karen Brodersen[1,3], Anne Hauch[1], Henrik Lund Frandsen[1], Theis Løye Skafte [1,4], Åsa H. Persson[1], Jens V. T. Høgh[1,3], Henrik Henriksen[1], Maria Navasa[1,5], Xing-Yuan Miao[1], Xanthi Georgolamprou[1], Søren P. V. Foghmoes[1,6], Peter Vang Hendriksen[1], Eva Ravn Nielsen[1,7], Jimmi Nielsen[1,8], Anders C. Wulff[1], Søren H. Jensen[1,9], Philipp Zielke[1,10] & Anke Hagen [1]✉

The transportation sector is undergoing a technology shift from internal combustion engines to electric motors powered by secondary Li-based batteries. However, the limited range and long charging times of Li-ion batteries still hinder widespread adoption. This aspect is particularly true in the case of heavy freight and long-range transportation, where solid oxide fuel cells (SOFCs) offer an attractive alternative as they can provide high-efficiency and flexible fuel choices. However, the SOFC technology is mainly used for stationary applications owing to the high operating temperature, low volumetric power density and specific power, and poor robustness towards thermal cycling and mechanical vibrations of conventional ceramic-based cells. Here, we present a metal-based monolithic fuel cell design to overcome these issues. Cost-effective and scalable manufacturing processes are employed for fabrication, and only a single heat treatment is required, as opposed to multiple thermal treatments in conventional SOFC production. The design is optimised through three-dimensional multiphysics modelling, nanoparticle infiltration, and corrosion-mitigating treatments. The monolithic fuel cell stack shows a power density of 5.6 kW/L, thus, demonstrating the potential of SOFC technology for transport applications.

[1] Department of Energy Conversion and Storage, Technical University of Denmark, Kgs, Lyngby, Denmark. [2] Present address: Department of Sustainable Energy Technology, SINTEF Industry, Oslo, Norway. [3] Present address: Haldor Topsoe A/S, Kgs, Lyngby, Denmark. [4] Present address: Noon Energy Inc., Palo Alto 94306 CA, USA. [5] Present address: Alfa Laval Lund AB, Energy Division, Lund, Sweden. [6] Present address: Esti Chem A/S, Gadstrup, Denmark. [7] Present address: Ramboll Group A/S, Copenhagen, Denmark. [8] Present address: Radiometer Medical ApS, Brønshøj, Denmark. [9] Present address: DynElectro ApS, Viby, Denmark. [10] Present address: FOSS A/S, Hillerød, Denmark. ✉email: stepir@dtu.dk; anke@dtu.dk

Societies worldwide are transforming their energy systems to gradually become independent of fossil fuels. The transport sector accounts for ca. 25% of the total energy consumption[1]. Fuel cell-powered electric vehicles and ships or use of fuel cells as range extenders help in reducing greenhouse gas emissions from transport. Fuel cells can convert the chemical energy of sustainable fuels produced from water electrolysis with zero-carbon electricity, such as green hydrogen, green ammonia, or green methane, directly into electricity with a high electrical efficiency, above 45%[2–4]. When hydrogen and ammonia are used as fuels, the emission of particles, such as $NO_x$, or $CO_2$ are near zero during power production[5,6]. Fuel cells provide long driving ranges, rapid re-fuelling[7], and efficient conversion of liquid e-fuels to power in ocean-going ships. However, to ensure the widespread use of fuel cells in transportation, they should exhibit high efficiency, low cost, high volumetric density and specific power, and sufficient durability.

Low-temperature fuel cells (proton-exchange membrane fuel cells (PEMFCs)) have achieved a certain degree of technological maturity for automotive applications[8–10]; several large automotive and bus manufacturing companies are producing such vehicles in small series (around 25,000 vehicles were in operation at the end of 2019)[11]. An advantage is a swift system start-up[12]. However, PEMFCs require high hydrogen purity[13] and are relatively expensive owing to the use of noble metals[14]. On the other hand, high-temperature fuel cells (solid oxide fuel cells (SOFCs)) are commonly used for stationary applications; their high operating temperatures (600–1000 °C) make them particularly efficient for combined heat and power systems[15,16]. They are also attractive for transport applications as noble metals are not required, and higher efficiencies of 60%[17] (compared to 45–50% for PEMFCs)[18,19] can be achieved. The high temperatures entail a slower start-up and thus also hints towards more constant operating conditions, such as in long-haul trucks, trains, or ships. The advantage is that they can be operated using less pure hydrogen, ammonia, methane, and other liquid fuels[20,21].

Although the first usage of SOFCs as range extenders in vehicles was demonstrated during the last decade[22,23], the technology was considered less suitable for vehicles because of the high operating temperature, limited thermal cycling robustness of the ceramic-based cells[24], and low power density (typical commercial SOFCs currently operate in the range of 0.1–1.0 kW/L)[22,25].

This study presents a novel concept for fabricating a metal-based monolithic, high-temperature fuel cell stack with high power density (5.6 kW/L) using cost-competitive and scalable manufacturing methods.

## Results

The concept of the metal-based monolithic stack is illustrated in Fig. 1 and compared to the conventional stacking of ceramic anode-supported fuel cell stacks with metallic interconnects. In a conventional stack (both metal and ceramic supported), the cell support, gas channels, and interconnects constitute the thickest layers (few hundred micrometres), while the active part of the cell is in the range of few tens of micrometres. The monolith concept integrates the cell support, gas channels, and the interconnects into a single layer, thereby reducing the stack height by a factor of 2–4. This considerably increases the volumetric density and specific power of the stack, which are important parameters considering the space and weight constraints in vehicles. More specifically, the power per volume is significantly improved, and values of 6–8 kW/L are feasible.

In addition to the advantage of high power density, the metal-based monolith stack is significantly cheaper than a conventional

stack. The volume reduction alone reduces the materials costs, and further reduction is achieved by using cost-effective metals[26] (Fe is much cheaper than Ni and zirconia ($ZrO_2$) used in the conventional stack[27]) for the thickest component of the monolith. The high metal content also reduces temperature differences across the stack during operation and provides greater robustness to fast thermal cycles. This is important for hybrid automotive applications integrating SOFCs and electrical battery systems[28].

Previous work on monolithic SOFCs designs were reported by Saint-Gobain, which developed fully ceramic monolithic SOFC stacks[29], and by Argonne National Laboratory, which designed metal-supported SOFC stacks ("TuffCell")[30]. However, our concept of metal-based monolithic SOFCs comprises several significant differences when compared to the one developed by Argonne National Laboratory, such as the processing techniques to integrate gas flow channels, the electrode materials and thicknesses, and the concept of integrated seals. Thus, our single repeating unit (SRU) monoliths are significantly thinner (>2 times thinner) and integrate cell components, interconnects, gas distribution channels, and seals in a single heat treatment step while the "TuffCell" incorporates seal and cathode only after co-sintering.

The following sections present the underlying concepts for monolith design, manufacturing methods, achieved module, and electrochemical performance of the SRU monolith.

**Concept for monolith stack design.** The schematic illustration of the monolith stack is shown in Fig. 1. The electrolyte, composed of scandia-doped yttria-stabilised zirconia (ScYSZ, 10 μm), is sandwiched between two porous electrodes (ScYSZ-Fe22Cr composite, 15 μm) with dense sealing strips (ScYSZ-Fe22Cr, 15 μm) along the sides. These are placed between thicker layers corresponding to gas distribution channels (250 μm) and metallic interconnects (dense Fe22Cr, 150 μm). The SRU monolith is co-sintered in one firing step, which results in more robust interfaces than those in conventional stack designs and reduces the total number of firing steps in the manufacturing process. The integrated seals placed between the metallic interconnects and the ceramic electrolyte are laterally joined to the cermet electrodes to ensure gas tightness and improved thermocycling stability. To avoid thermal expansion coefficient mismatches with the surrounding components, while maintaining the electronic insulating property required for SOFC applications, the seals are made of a 5:1 volume ratio of ceramic to metal.

The main design advantage of the monolith is the integration of gas flow channels into a co-sintered block containing both a metallic substrate and ceramic cell interlayers. This results in high power density and improved in-plane thermal distribution. The gas channels are placed between the interconnects and the electrodes to efficiently feed gases to the electrodes and minimise the volume of the cell. They are formed via honeycomb pore-former tapes containing exclusively organic materials that burn off during the firing step.

The gas channel dimensions ($250 \times 500$ μm$^2$) are studied via a 3D multiphysics model built in COMSOL Multiphysics and optimised to obtain a tolerable pressure drop. In this model, the overall conservation laws are solved on a homogenised volume, and the microstructural details are introduced through effective material parameters. The model includes current, heat, and mass transport, including gas flows in channels, and finally, an assessment of the mechanical stresses in the structure.

An example of the model simulation is shown in Supplementary Fig. 1. Here, the calculated relation between the gas overblow, pressure drop, and inlet gas temperature, which ensures that the stack temperature never exceeds 700 °C, is reproduced when the

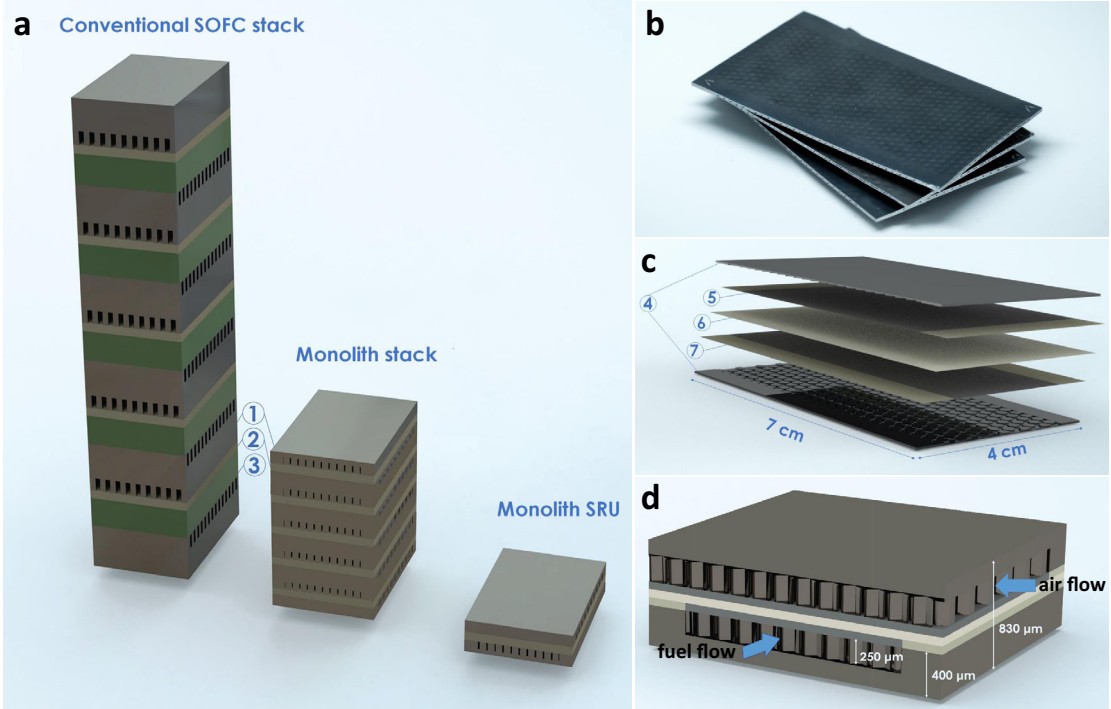

**Fig. 1 Metal-based monolithic stack design. a** Illustrations of conventional and monolith fuel cell stacks with five repeat units and a single repeating unit (SRU) monolith stack. The numbers 1, 2, and 3 correspond to the interconnects, fuel cells (electrolyte and electrodes), and cell supports, respectively. **b** Photograph of three modules of SRU monolith. **c** Illustration of an exploded view of a SRU monolith. The numbers 4, 5, 6, and 7 correspond to interconnects and gas distribution channels (400 μm thick including 250 μm high gas channels), electrodes (10 μm), electrolyte (10 μm), and sealing (10 μm), respectively. **d** Enlarged cross-sectional view of a SRU monolith (for clarity, the scale of the layers thickness is not respected).

stack is operated at 0.67 A/cm². The calculations confirm that cooling of the monolith stack is feasible with achievable gas channel dimensions (250 × 500 μm²) and a maximum pressure drop of 300 mbar. Note that the catalyst infiltration was not accounted for in the simulation of the pressure drops as infiltration primarily influences the diffusion in the dense microstructure, but not the convection in the channels. This slight overpressure of airflow could also be beneficial to handle the heat production arising from the specific compactness of the monolith stack design, especially once integrated in a SOFC system. This challenge may also be solved by segmenting the monolith unit, with separating layers allowing for efficient heat transport, or designing operation strategies to optimize the heat production.

**Manufacturing**. The monolith is fabricated using tape casting, lamination, co-sintering, and catalyst infiltration, which are well-known methods for the manufacture of high-temperature fuel cells[31,32].

The manufacturing of monolithic stacks is challenging as it requires the sintering of various ceramic and metallic particles while maintaining a hierarchical microstructure for the main gas flows and gas diffusion paths. One of the most complex steps is the binder burnout of these large and compact monolithic cells. This challenge may limit the number of repeating units in a monolith stack and require segmenting the assembly.

To identify the process parameters required to achieve an optimized monolith (avoiding disintegration and warpage during debinding), a 3D model simulating the debinding process was developed using COMSOL Multiphysics. The model describes the pressure inside the hierarchical microstructure emerging as the organics are burned. Kinetic models for the combustion of

organics were developed and calibrated by fitting the measured weight change in thermo gravimetric analysis to the simulated ones. These are combined with a description of convection in the gradually changing microstructure as the organics burn. The mass transport occurs in the hierarchical microstructure with features ranging from micrometres to several centimetres. To effectively describe this, a multi-scale model was employed. Here, the diffusion of decomposed organics in the porous microstructures are described by Fick's diffusion, whereas the flow in the channels being formed is described by the Darcy-Weissbach equation, which can be simplified to Darcy's law, when the structure is homogenously distributed in the considered domain[33,34]. The latter approach allows for describing both the diffusion and the convection at different length scales in the same computational homogeneous media[33,34]. The heat transport is both through convection and conduction. The reaction kinetics are described through a set of Arrhenius expressions and add to the mass balances in the diffusion equation.

The model was used to identify the optimal heating ramps and the composition of the organics to minimise the overpressure inside the microstructure. For example, the calculations indicate that using a graphite/poly(methyl methacrylate) (PMMA) mixture in the sacrificial "pore-former" tapes is beneficial compared to using only PMMA, as the decomposition is spread over a larger temperature span. Figure 2 shows SRU monoliths manufactured with different gas channel forming materials (Fig. 2a 100% graphite, Fig. 2b 50–50% graphite–PMMA, and Fig. 2c 100% PMMA) in addition to the organics use in the tapes. The model describes how the burning of the different pore-forming agents increases the pressure generated in the monolith while using fixed gas channel dimensions and heating profile. Photographs of post-sintered SRU monoliths manufactured using the different pore-former agents to form gas channels are

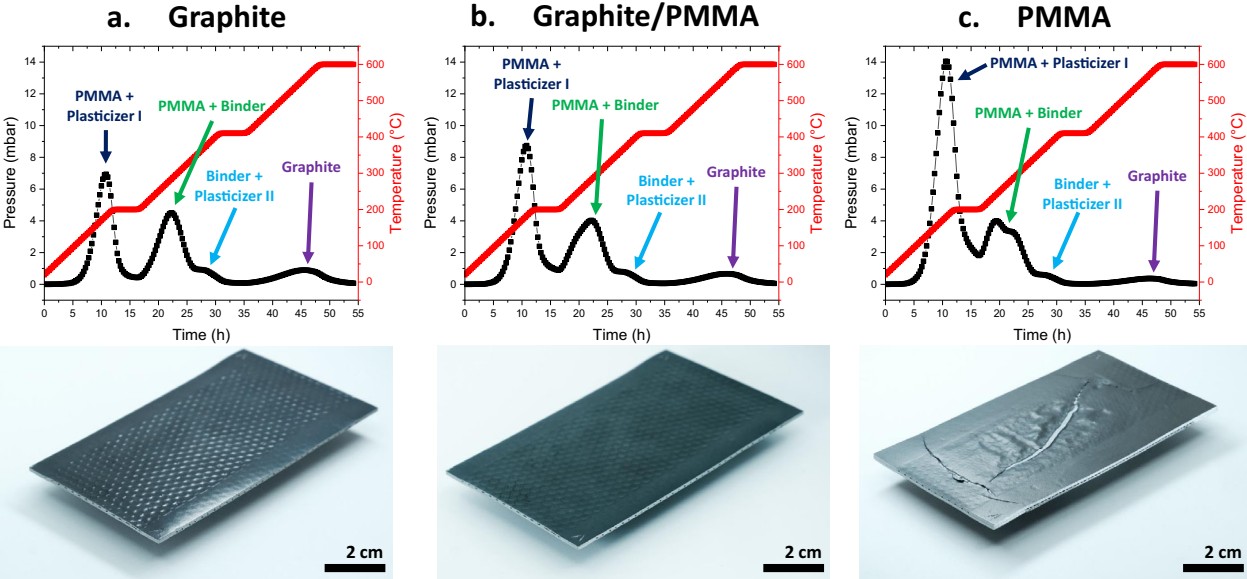

**Fig. 2 Effect of pore-forming materials on built-up pressure during the debinding step.** The graphs display the predictions of a multiphysics model on the pressure built-up inside the SRU monoliths (black lines/symbols) during the debinding step (red lines). Below the graphs, the photographs show the integrity of the corresponding SRU monoliths after heat treatment (debinding and sintering steps). **a**, Case of a SRU monolith manufactured using only graphite as sacrificial material to form the gas channels, **b**, Case of a SRU monolith manufactured using a 50–50 vol.% mixture of graphite–PMMA as sacrificial material to form the gas channels, and **c**, Case of a SRU monolith manufactured using only PMMA as sacrificial material to form the gas channels. The SRU presented in Fig. 2c reveals large cracks after debinding/sintering steps which is in good accordance with the model which predicted that SRU monolith manufactured using 100% PMMA would lead to the highest pressure (14 mbar around 200 °C) among the three pore-forming agents investigated, and therefore will be the most likely to fracture. Note that both PMMA and graphite are also contained the electrode tapes which explains why a pressure peak corresponding to graphite removal can also be found in the case of Fig. 2c, for example.

presented in Fig. 2 below the graphs. The monolith made from PMMA exhibited cracks. This is in good accordance with the model, which predicted that the use of 100% PMMA would lead to the highest pressure (14 mbar around 200 °C) among the three pore-forming agents investigated.

Another difficult task is the co-sintering of the SRU monolith. The sintering shrinkage of the different layers must be matched to ensure that the cell structure withstands unavoidable mechanical stresses and prevents cracking of the monolith. Hence, the shrinkage of each single layer was investigated by dilatometry and adjusted by optimising the particle size distribution, composition, shape, and ratio of raw materials and organic additives. Complex casting techniques, such as co-casting and side-by-side casting, were used to improve the adhesion of the layers. The composition of the FeCr/ScYSZ electrodes and seals was tuned to match the macroscopic thermal expansion of these layers with that of the metal support and the electrolyte (see Supplementary Fig. 2). All these process optimisations were required to obtain flat and crack-free monoliths as shown in the photographic pictures of Fig. 1 and scanning electron microscopy (SEM) images of Fig. 3a and Fig. 3b.

**Fuel cell functionalisation**. The electrochemical activity was introduced by infiltrating various suitable nitrate solutions, that is Gd-doped ceria without (CGO) and with nickel (Ni-CGO), and lanthanum strontium cobalt (LSC), on the fuel and oxygen sides, respectively. Stoichiometric amounts of the required nitrates were mixed with deionised water and surfactant (Pluronic P123) to obtain a concentration of 1–2 M. Because the electrodes in the monolithic design do not have exposed surfaces, the nitrates must be injected through the gas channels and spread to the porous electrode structure. This was accomplished using a clamping device (see Supplementary Fig. 3) that can hold the individual monolith (one SRU or an entire stack) in place, while a slight pressure

pushes nitrates through the open gas channels. Subsequently, the monolith was heated to 325 °C[35,36] at 2 °C/min to evaporate water, decompose nitrate salts, and form the desired phase (see Supplementary Fig. 4 for thermogravimetric analysis). The procedure was repeated to achieve sufficient coverage; this ensures electrochemical activity and corrosion protection. The energy dispersive X-ray spectroscopy (EDS) mapping image with elemental distribution analysis of a typical coverage after three infiltration cycles is shown in Supplementary Fig. 5. The weight gain of the SRU monolith increased after each infiltration cycle and plateaued after approximately 10 cycles (see Supplementary Fig. 6).

**Corrosion stability**. In addition to providing electrochemical activity, the second important function of the CGO layer is to provide corrosion protection of the metallic phase in the electrode backbone, gas distribution layer, and the interconnect. On the fuel side, the most corrosive location is near the gas outlet (highest water vapour pressure). To increase the overall system efficiency, high fuel utilisation is required; outlet gas compositions containing less than 20% $H_2$ and more than 80% steam are relevant operating conditions, as reported previously[37]. The corrosion rate on the air side is lower than that in steam, considering the relatively low target operating temperature (650 °C)[38–40]. Nevertheless, a protective layer is needed to prevent evaporation of Cr from the metal as it can poison the air electrode. It remains to be tested whether the CGO layer can provide sufficient protection against Cr evaporation. As far as we are aware, the only measurement of Cr evaporation from steel with a Ce-based coating was performed by Grolig et al.[41]. The study shows that a 10 nm thin Ce layer does not reduce Cr evaporation from AISI 441 steel. However, this result is not directly comparable to the coating applied in our work, which is thicker and contains Gd in addition to Ce. In case the CGO layer is found to be insufficiently protective, there are other well-proven coatings

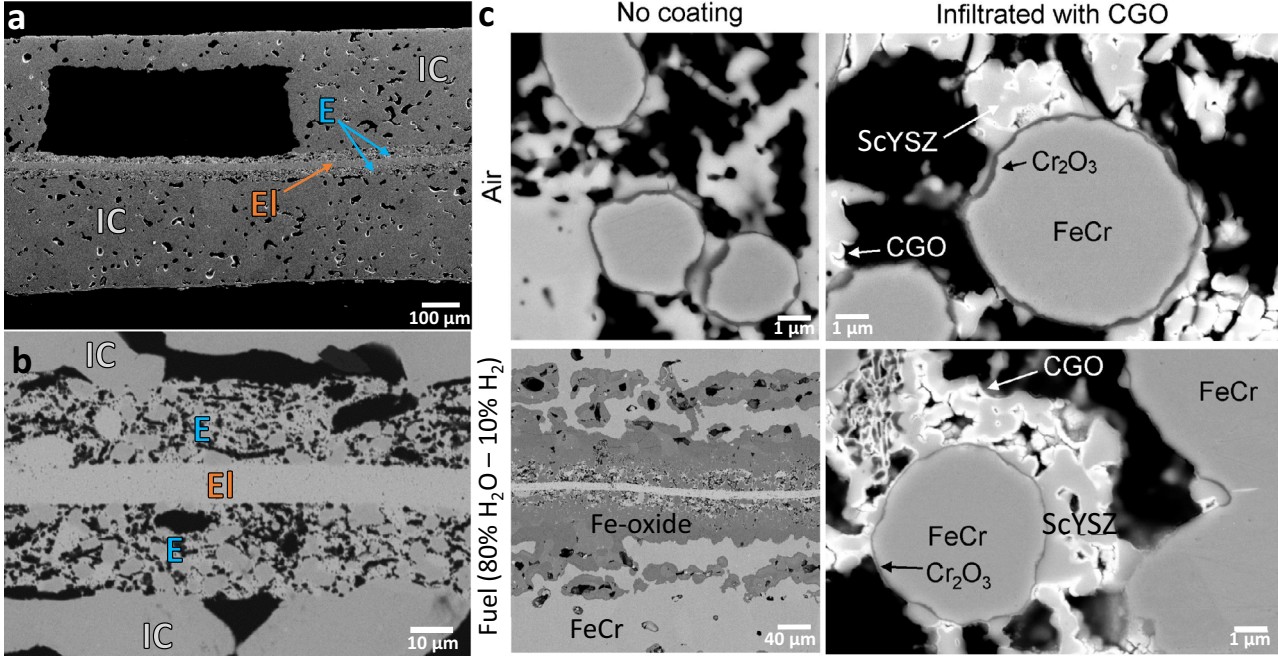

**Fig. 3 Microstructural analyses of the monolith. a** Cross-sectional SEM image showing the SRU monolith with a gas channel enlarged. The annotations E, El, and IC correspond to electrode, electrolyte, and interconnect, respectively. **b** High-magnification cross-sectional SEM images of the cell layers. **c** Cross-sectional SEM images of the SRU monoliths after oxidation in air or 80% $H_2O$–20% $H_2$ for 100 h at 650 °C. Note the scale difference.

that can be applied on the air side instead, for example $MnCo_2O_4$[42].

Figure 3c shows the cross-sectional SEM images of the SRU monolith after oxidation in air (ambient $H_2O$ level) and 80% $H_2O$–20% $H_2$ for 100 h at 650 °C (see Supplementary Fig. 7 for SEM images with matching scale). This temperature was chosen as it corresponds to a critical temperature in terms of break-away oxidation in fuel side atmospheres with a high $H_2O$ content. At higher temperatures, $Cr_2O_3$ is more easily formed, protecting the steel against Fe-oxide formation[43]. From the images of the non-coated monolith, it is evident that corrosion protection is most critical at the fuel side of the monolith. After oxidation in air, the metal is covered by a thin Cr-rich layer (most likely $Cr_2O_3$), while after oxidation in the fuel side atmosphere, a large amount of non-protective iron oxide is formed. By infiltrating the monolith once with CGO and pre-oxidising the monolith at 850 °C in $H_2$–$N_2$ to densify the layer, oxidation was strongly mitigated. However, some areas of the electrode backbone exposed to the fuel atmosphere showed signs of iron oxide formation. This is likely due to insufficient coverage by the CGO layer. The coverage improved after three rounds of infiltration, and no iron oxide formation was observed (see Supplementary Fig. 8). Pre-oxidation alone had a minor role in reducing the detrimental oxidation (see Supplementary Fig. 8). Due to the small particle size (5–10 μm) of the steel in the electrode layers, there is a limited reservoir of Cr available to maintain the $Cr_2O_3$ scale[44]. Although the CGO layer provided sufficient protection to prevent Cr depletion within 100 h, tests of longer duration (1000 s of h) are needed to confirm that the coating sufficiently slows down the oxidation rate to allow the use of such small metal particles.

**Electrochemical performance**. A special test fixture was developed to test the SRU monolith, as described in the Supplementary Fig. 9. Figure 4a shows the initial electrochemical performance of the SRU monolith cell at 780 °C. An open circuit voltage (OCV) of 1065 mV was measured. Under the given test conditions, this corresponds to ~1.75% of the $O_2$ at the oxygen electrode leaking

to the fuel electrode side, assuming that the lower OCV compared to the theoretical Nernst potential is solely due to leakage from the oxygen side of the cell. From a fundamental diffusivity point of view hydrogen crossover will be more likely. However, based on the observations made on the tested cells (i.e., the presence of pin holes with clear corrosion problems) and the fact that the OCV is lower than the theoretical Nernst potential, it is more likely an oxidizing gas that crosses over and not the hydrogen. Note that, prior to characterisation, the SRU monoliths were leak-tested using a set-up developed in-house (see Supplementary Fig. 10), thus ensuring that the interconnect layers are sufficiently dense. The leak could possibly have originated from the test fixture and not the monolith itself. Nonetheless, the measured OCV is considered acceptable for a cell in the early stage of development. However, electrolyte gas tightness and sealing need to be further improved.

The tested SRU monolith had an active cell area of ~18 cm²; based on the i–V curve reported in Fig. 4a, this can be translated to an area-specific-resistance of 0.49 Ω cm² for data from OCV to 0.6 A/cm². Note that at a current density of ~0.67 A/cm², the course of the i–V curve suggests that mass transport and/or fuel starvation negatively affect the performance of the SRU monolith. Assuming a cross-over leak of 1.75% of oxygen to the fuel electrode, the $H_2$ utilisation was 24% at 0.67 A/cm² indicating scope for improvement with regard to the $H_2$ feed and distribution to the active sites.

Figure 4b shows the comparison of the initial electrochemical performances of the SRU monolith and the state of the art (SoA) commercially available Ni/YSZ anode supported ceramic SOFCs described in the literature[45–47]. The proposed compact monolith SOFC design is attractive for mobile applications where power per volume is an important parameter. Figure 3b compare the height of the SRU monolith design developed in this work with those of common commercially available stacks, spanning from 1.4 mm to 4 mm. The SRU monolith achieved a value of 5.6 kW/L, on a par with the best performing SoA Ni/YSZ anode supported ceramic SOFCs which have been optimised over decades.

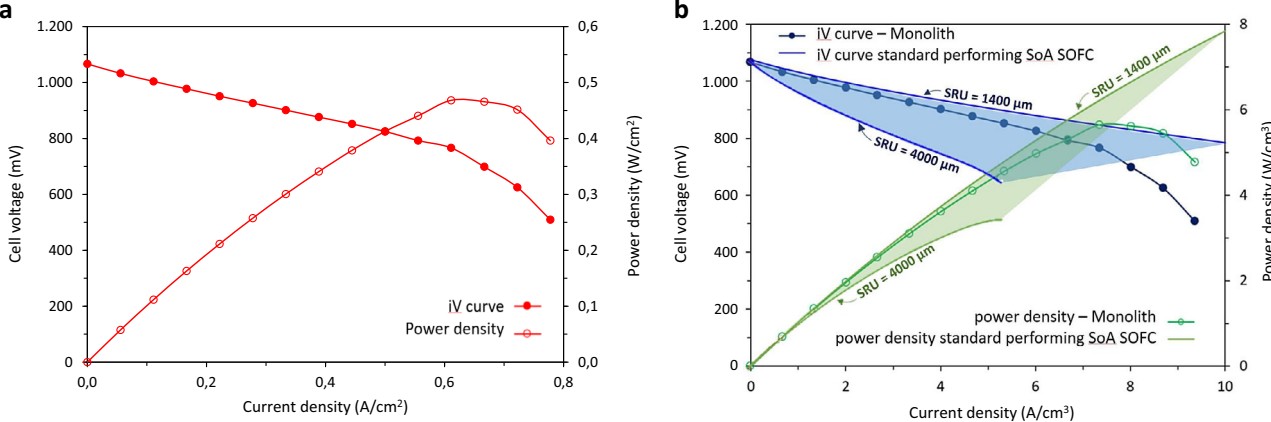

**Fig. 4 Electrochemical performance and comparison with the SoA SOFCs. a** i–V curve (full symbol) and power density (empty symbol) of SRU monolith cell with an active cell area of ∼18 cm$^2$ measured at 780 °C, dry H$_2$ (25 l/h) to the fuel electrode, and O$_2$ to the oxygen electrode. **b**, Comparison of initial electrochemical performance of the SRU monolith cell (blue full symbol) with that of SoA commercially available Ni/YSZ anode supported SOFCs (green empty symbol). The power density in terms of power per volume is based on a SRU height of 830 μm for the monolith and in the range of 1400–4000 μm for the SoA anode supported SOFCs design.

To conclude, a novel metal-based monolithic SOFC was fabricated using cost-effective and scalable manufacturing techniques. Initial electrochemical tests revealed a high power density (5.6 kW/L), which indicates their potential for mobile applications. We speculate that this design, or similar, could enable large-scale production of electrified transportation powered by fuel cells with considerably increased range, decreased charging times, and lower cost.

## Methods

**Manufacturing of one repeating unit of monolith**. All the components of the monolith were fabricated via tape casting. Initially, a suspension composed of ceramic (ScYSZ, Daiichi Kigenso Kagaku Kogyo Co., LTD.) and/or metallic (Fe22Cr (21.4% Cr, 1.0% Mo, 0.2% Mn, <0.1% Si balanced with Fe), Sandvik Osprey LTD.) powder(s), solvent, and polymers (binder, plasticizers, dispersant) was homogenised in air for 4 days by ball milling (EB Kuglemølle Rack, EB Teknik Borup A/S, Denmark) using polyethylene bottles from 250 ml to 2 l and zirconia balls. The electrode suspensions were made of 60 vol.% ScYSZ and 40 vol.% Fe22Cr and also contained graphite and PMMA as pore former. The seals were composed of 80 vol.% ScYSZ and 20 vol% Fe22Cr. Prior to tape casting, the suspensions were degassed under vacuum to completely eliminate trapped air bubbles.

The seal/electrode/seal structure was fabricated first via side-by-side casting using a doctor blade trough, separated in compartments where the suspensions could be poured without mixing. Then, a pore-former tape previously cast and laser-cut into a honeycomb shape was laminated onto the electrodes using heated rolls in a double-roll set up. The interconnects were co-cast on top of the laminated structure. The complete SRU monolith was obtained by laminating a ScYSZ electrolyte between two blocks of interconnect/pore former/seal-electrode-seal rotated by 90°. The lamination was performed via an isostatic press at 85 °C under a pressure of 30 MPa. The green monoliths were debinded at 600 °C for 4 h in air and then sintered at 1290 °C for 6 h in H$_2$.

**Monolith infiltration**. A 2 M CGO precursor solution with 0.8:0.2 molar ratio was obtained by dissolving stoichiometric amounts of Ce(NO$_3$)$_3$ 6H$_2$O and Gd(NO$_3$)$_3$ · xH$_2$O. Similarly, a 1 M Ni-CGO precursor solution was prepared from Ni(NO$_3$)$_2$ 6H$_2$O, and a 1 M La$_{0.588}$Sr$_{0.392}$CoO$_{3-\delta}$ precursor solution from La(NO$_3$)$_3$ 6H$_2$O, Sr(NO$_3$)$_2$ 6H$_2$O, and Co(NO$_3$)$_2$ 6H$_2$O. Pluronic P123 was added at a concentration of 1 g per 100 mL. The cells were first infiltrated twice with CGO on both the fuel and oxygen electrode sides to form a thin ceramic barrier layer on steel, thereby mitigating corrosion during operation. Following each infiltration cycle of CGO, the monoliths were heated to 325 °C at a rate of 2 °C/min in stagnant air and held for 0.25 h before cooling. This evaporates the water, decomposes the nitrate salts, and forms the desired fluorite phase[35,36]. After the first infiltration cycle of CGO, the monolith cells were pre-oxidized by heating to 850 °C at a rate of 2 °C/min and holding for 2 h in 5% H$_2$/N$_2$. This is an additional step used to mitigate subsequent corrosion of Fe22Cr steel[48] by forming a dense chromia layer; it has the advantage of partially densifying the infiltrated CGO layer as well. Subsequently, the fuel and oxygen electrodes were infiltrated with one cycle of Ni-CGO and LSC, respectively and heat treated in air at 400 °C for 0.5 h.

**Corrosion study**. To investigate the oxidation resistance, the sintered monoliths infiltrated with 0, 1, or 3 rounds of CGO were cut into ca. 2 × 2 cm$^2$ pieces using a diamond blade. The pieces were placed on an alumina support and heat treated in air (ambient H$_2$O content, ca. 2–4%) or 80% H$_2$O–20% H$_2$ for 100 h at 650 °C using a heating and cooling rate of 120 °C/h. An 80% H$_2$O–20% H$_2$ atmosphere was achieved by mixing H$_2$ and O$_2$ at the furnace inlet, which are directly combusted owing to the high temperature. After oxidation, the monolith pieces were cast in epoxy and polished to reveal the cross sections.

**Characterisation techniques**. Optical dilatometry was used to determine the shrinkage characteristics of the interconnect, electrode, seal, and electrolyte tapes during both the debinding and sintering steps. To study the debinding step, the measurements were carried out in a TOMMI optical dilatometer (Fraunhofer Institut Silicatforschung ISC, Germany) in air using green tapes rolled into a cylinder (5 mm diameter, 10 mm length). To study the sintering step, the measurements were carried out in a graphite heated thermo-optical measurement (TOM) device (model: TOM_metal, from Fraunhofer-Center for High Temperature Materials and Design, Germany). The measurements were performed in a reducing atmosphere using previously debinded tapes rolled into cylinders.

The fractured and polished cross-sectional microstructures of the as-prepared and the tested monoliths were investigated by scanning electron microscopy (SEM) using a Zeiss Merlin scanning electron microscope equipped with a field emission gun and a Hitachi TM3000 equipped with a Bruker energy dispersive X-ray spectroscopy (EDS) system.

**Electrochemical performance tests**. The single-cell monoliths were tested in a commercially available FuelCon Evaluator test rig. A metallic test house was constructed for these cells. The metallic test house enables fast heating and cooling. The monolith was placed in the metallic test house using mica sheets for external sealing. Gas tightness was achieved by joining the two parts of the cell house with screws. The voltage and the current probes were placed equidistantly from the sides of the cell. The fuel inlet/outlet was located on the long side of the monolith, while the air inlet/outlet was placed perpendicularly on the short side to ensure crossflow, as shown in the design of the monolith (see Supplementary Fig. 9).

The monolith was heated to 650 °C at 10 °C/h, feeding 5% H$_2$ balanced with N$_2$ (10 $_N$L/h) to the fuel electrode and air (10 $_N$L/h) to the oxygen electrode. At 650 °C, 5% H$_2$/N$_2$ was replaced with pure H$_2$ (10 $_N$L/h). The cell voltage was measured during start-up and throughout the cell tests. The initial electrochemical performance of the SRU monolith cell was measured at 780 °C, feeding pure H$_2$ (25 $_N$L/h) to the fuel electrode and O$_2$ (100 $_N$L/h) or air (100 $_N$L/h, see Supplementary Fig. 11) to the oxygen electrode. Based on few examples/numbers from literature[49,50] SRU height in the range from 1.4 mm to 4 mm is reasonable to assume for SoA SRU and these numbers were applied to compare the electrochemical performances of the monolith with SoA SOFCs in terms of power density per volume, as presented in Fig. 4.

## Data availability

The data generated in this study have been deposited in a Figshare database [https://figshare.com/s/9b510011e411fda46a7d].

## Code availability

The codes that support the findings of this study are available on demand from the corresponding authors.

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

## Acknowledgements

The authors gratefully acknowledge financial support from Plastic Omnium. We are also thankful to Henrik Paulsen for assistance in preparing samples for SEM/EDS analyses. We thank Enzo Moretti, Ilaria Ritucci, and Ragnar Kiebach for improving the images of the monoliths. We are grateful to Søren Christensen for assisting with the preparation of slurries for tape casting.

## Author contributions

S.P., B.T., K.B., T.L.S. and A.Hau. designed the experiments leading to the final monolith design. S.P. and K.B. manufactured the samples. S.P., K.B., A.H.P. and S.P.V.F. carried out preliminary experiments on manufacturing. T.L.S., S.P. and B.T. contributed to the fuel cell functionalization. B.T. contributed to the corrosion study. S.P. and B.T. performed the microscopy analyses. A.Hau., J.V.T.H. and H.H. conducted electrochemical tests. M.N., H.L.F. and X-Y.M. contributed to the modelling studies. A. Hag., E.R.N., B.T., J.N. and P.V.H. supervised the research. J.N, A.C.W., S.H.J., P.Z., H.L.F. and A.Hag. contributed to

the initial monolith design. X.G. and S.P. contributed to the design of the figures. S.P., B.T., A.Hau., T.L.S., A.Hag. and H. L. F. contributed to writing the manuscript. A.H.P, M.N., X.G., P.V.H., J.N., A.C.W., S.H.J. and P.Z. revised the manuscript.

## Competing interests

The authors declare no competing interests.
