## [Peer Review File · Nature Communications]

REVIEWER COMMENTS

Reviewer #2 (Remarks to the Author):

The reviewer's comments and requests have been addressed adequately.

Reviewer #3 (Remarks to the Author):

The manuscript was first submitted to Nature Energy and has been reviewed by three reviewers. The authors were encouraged to transfer the manuscript to Nature Communications after considering the three reviews.

General comment: Metal supported fuel cells are highly attractive for the transportation sector due to their large potential with respect to thermal management (high heating rates) and robustness. The manuscript describes a method to produce a new type of metal supported SOFC stack, which is characterized by its assembling as a monolith. Based on electrochemical tests done on a single repeat unit (SRU, size of 70 x 40 mm²), the authors state a power density of 5.6 kW/L for a monolithic stack made of these SRUs. This power density clearly exceeds the requirements of SOFC stacks in the transportation sector, e.g. as range extender for battery electric vehicles. Furthermore, the authors present a production route for this monolithic stack, which is optimized with respect to cost effectiveness amongst others by reducing the thermal treatments during cell processing. The results are promising and of interest for the SOC community.

The authors accurately considered the recommendations of the reviewers and changed the manuscript accordingly. Also further supplementary information is added for clarification.

I have only a few final comments to the revised manuscript:

- Figure 2a: I wonder why there are pressure peaks, which are related to PMMA, even if pure graphite is used as pore-forming material in this case.
- Figure 3c: The small image of the oxidised microstructure with the 40 µm scale bar might be rotated by 90°, then this image would fit better to Fig. 3a,b.
- Page 15, lines 235 – 236 and Figure 4b: Could you specify more clearly, who are the suppliers of the ASC-SRUs used for the benchmark. Please add this information in the "methods" part.
- Reference 21: Not clear if journal publication or patent, please revise.

Point-by-point response letter to the reviewers

The authors would like to thank the reviewers for their time and their useful comments, helping to improve this paper. All comments from the reviewers were answered and the manuscript was modified accordingly. Below the detailed reply to each issue addressed by the reviewers can be found.

The comments from the reviewers are written in red, the answers and comments from the authors are marked in blue. The modifications made in the manuscript are highlighted in yellow.

REVIEWER COMMENTS

Reviewer #2 (Remarks to the Author):

The reviewer's comments and requests have been addressed adequately.

Reviewer #3 (Remarks to the Author):

The manuscript was first submitted to Nature Energy and has been reviewed by three reviewers. The authors were encouraged to transfer the manuscript to Nature Communications after considering the three reviews.

General comment: Metal supported fuel cells are highly attractive for the transportation sector due to their large potential with respect to thermal management (high heating rates) and robustness. The manuscript describes a method to produce a new type of metal supported SOFC stack, which is characterized by its assembling as a monolith. Based on electrochemical tests done on a single repeat unit (SRU, size of 70 x 40 mm²), the authors state a power density of 5.6 kW/L for a monolithic stack made of these SRUs. This power density clearly exceeds the requirements of SOFC stacks in the transportation sector, e.g. as range extender for battery electric vehicles. Furthermore, the authors present a production route for this monolithic stack, which is optimized with respect to cost effectiveness amongst others by reducing the thermal treatments during cell processing. The results are promising and of interest for the SOC community.

The authors accurately considered the recommendations of the reviewers and changed the manuscript accordingly. Also further supplementary information is added for clarification.

I have only a few final comments to the revised manuscript:

- Figure 2a: I wonder why there are pressure peaks, which are related to PMMA, even if pure graphite is used as pore-forming material in this case.

There are pressure peaks related to PMMA in all cases because PMMA is also contained in the electrode tapes (as pore-forming agent). The experiment was performed on SRU monoliths therefore even the monolith containing only graphite as sacrificial material to form gas distribution channels (figure 2.a) contained some PMMA.

To clarify, we have modified a sentence in the manuscript (page 9, lines 10-12): *“Figure 2 shows SRU monoliths manufactured with different gas channel forming materials (a. 100 % graphite, b. 50–50 % graphite–PMMA, and c. 100 % PMMA) in addition to the organics use in the tapes.”*

Figure 3c: The small image of the oxidised microstructure with the 40 μm scale bar might be rotated by 90°, then this image would fit better to Fig. 3a,b.

Figure 3 has been modified accordingly.

- Page 15, lines 235 – 236 and Figure 4b: Could you specify more clearly, who are the suppliers of the ASC-SRUs used for the benchmark. Please add this information in the “methods” part.

A sentence and two references were added to the method section to specify how the ASC-SRUs benchmark was estimated:

“The monolith was heated to 650 °C at 10 °C/h, feeding 5 % H₂ balanced with N₂ (10 nL/h) to the fuel electrode and air (10 nL/h) to the oxygen electrode. At 650 °C, 5% H₂/N₂ was replaced with pure H₂ (10 nL/h). The cell voltage was measured during start-up and throughout the cell tests. The initial electrochemical performance of the SRU monolith cell was measured at 780 °C, feeding pure H₂ (25 nL/h) to the fuel electrode and O₂ (100 nL/h) or air (100 nL/h, see supplementary Fig. 11) to the oxygen electrode. Based on few examples/numbers from literature^{41,42} SRU height in the range from 1.4 to 4 mm is reasonable to assume for SoA SRU and these numbers were applied to compare the electrochemical performances of the monolith with SoA SOFCs in terms of power density per volume, as presented in Figure 4.”

- Reference 21: Not clear if journal publication or patent, please revise.

Reference 21 is a patent. The reference has been revised and the patent number has been specified in the manuscript (page 22).